**Data Availability Statement:** The minimum dataset has been deposited on the Open Science

# Effect of poor glycemic control on the prevalence and determinants of anemia and chronic kidney disease among type 2 diabetes mellitus patients in Jordan: An observational cross-sectional study

**Ahmed Al-Dwairi**[1]*, **Othman Al-Shboul**[1], **Doa'a G. F. Al-U'datt**[1], **Rami Saadeh**[2], **Mohammad AlQudah**[1,3], **Adi Khassawneh**[2], **Mahmoud Alfaqih**[1,4], **Alhakam Albtoush**[5], **Aysam Hweidi**[5], **Abdulaziz Alnemer**[5]

1 Faculty of Medicine, Department of Physiology and Biochemistry, Jordan University of Science and Technology, Irbid, Jordan, 2 Faculty of Medicine, Department of Public Health and Community Medicine, Jordan University of Science & Technology, Irbid, Jordan, 3 Department of Physiology, College of Medicine and Medical Sciences, Arabian Gulf University, Manama, Bahrain, 4 Department of Biochemistry, College of Medicine and Medical Sciences, Arabian Gulf University, Manama, Bahrain, 5 Faculty of Medicine, Jordan University of Science & Technology, Irbid, Jordan

* Andwairi7@just.edu.jo

## Abstract

### Background and objectives

Anemia and chronic kidney disease (CKD) are common findings in diabetic patients. Lack of glycemic control is associated with increased risk of diabetic complications. This study aimed to determine the effect of poor glycemic control on the prevalence and determinants of anemia and CKD among type 2 diabetes mellitus (T2DM) patients in Jordan.

### Methods

A cross-sectional study design was used in this research. T2DM patients with controlled diabetes (HbA1c $\leq$7.0%, n = 120) and age-, gender- and body mass index–matched uncontrolled diabetic patients (HbA1c >7.0%, n = 120) were recruited. Blood sample for $HbA_{1c}$ and serum insulin measurement were obtained. Complete blood count and kidney function test results were obtained from the patient's medical records. Anemia was determined according to World Health Organization criteria. A binomial logistic regression was performed to ascertain the effects of age, gender, CKD and glycemic control on the likelihood that participants have anemia.

### Results

The prevalence of anemia was significantly higher in the uncontrolled T2DM compared to controlled T2DM patients (40% vs 27.5%, OR: 2.14, 95% CI: 1.23, 3.71, P = 0.006). Female patients with uncontrolled T2DM had significantly greater prevalence of anemia compared to male patients with uncontrolled T2DM. The binomial logistic regression analysis showed

Framework platform and can be accessed via the following link: https://doi.org/10.17605/OSF.IO/T3FYB.

**Funding:** This work was supported by the Deanship of Research at Jordan University of Science and Technology (Grant # 20190009).

**Competing interests:** The authors have declared that no competing interests exist.

that age, female gender, and CKD were positively associated with anemia in the multivariate model, while in the univariate model, lack of glycemic control increases the odds of anemia by 1.74 (95% CI: 1.01, 2.99, P = 0.046).

## Conclusion

Anemia is commonly present among T2DM patients in Jordan and is associated with poor glycemic control especially in females. These results emphasize the necessity of including anemia screening in standard diabetes care to enable early detection and treatment of anemia and to enhance the overall care of diabetic patients.

## Introduction

Diabetes Mellitus (DM) is considered a global public health emergency that affects more than 537 million adults worldwide (prevalence of 10.5%) and linked to about 6.7 million deaths in 2021 in individuals aged 20–79 years [1]. In the Middle East and North Africa (MENA) region, 54.8 million adults live with DM (prevalence of 12.8%), and this is considered the second highest DM prevalence worldwide, with a projected increase to 107.6 million cases by 2045 [2]. Type 2 DM (T2DM) which accounts for approximately 90–95% of all DM cases in the world, is caused by insulin resistance in peripheral target tissues and relative insulin insufficiency [3]. Of note, the worldwide prevalence of DM has steadily increased in the past two decades primarily due to increased T2DM prevalence [1]. The middle-to-low-income countries, which are home to 80% of diabetic patients according to the International Diabetes Federation (IDF), are suffering from rapid increase in the incidence of T2DM [4–6].

T2DM is considered as a progressive multisystem disease characterized by disordered glucoregulatory hormones, that can lead to various micro- and macro-vascular complications [3,7,8]. Diabetic kidney disease is a common complication of poorly controlled T2DM, and is usually associated or preceded by development of anemia [9]. Anemia usually manifests earlier in diabetic patients with chronic kidney disease (CKD), and it is more severe in those with non-diabetic CKD [10,11]. Notably, anemia accelerates the development and progression of diabetic complications and is currently recognized as a risk factor for cardiovascular disease in diabetic patients [12,13]. Importantly, anemia is usually overlooked and undertreated due to lack of hematological screening in the primary care clinics [11,14].

According to the American Diabetes Association's (ADA) guidelines, the glycated hemoglobin (HbA1c) level should not exceed 7% [15] and T2DM is classified into controlled T2DM where the HbA1c ⬚ 7%, and uncontrolled T2DM where the HbA1c level exceeds 7% [15]. Importantly, large percentage of diabetic patients are unable to control their blood glucose levels despite treatment with different glucose-lowering medications, making them more vulnerable to development of diabetic complications such as anemia and CKD [16–21].

Sexual dimorphism may play a key role in the pathogenesis and outcomes of both anemia and CKD between men and women with T2DM [22]. Epidemiological studies indicate that anemia and CKD prevalence and incidence differs by sex, affecting more women than men, however, men are more likely to reach end stage renal disease than women [23–25].

The aim of this research was to identify the prevalence of anemia and CKD among T2DM patients, and to study the effect of glycemic control and gender on these conditions since their burden in diabetic patients has received little attention in Jordan. This research may pave the way to achieve better therapeutic outcomes for diabetic patients in the country.

## Materials and methods

### Study design

An observational case-control design was used in this research. The research received the ethical approval to recruit subjects to participate in the study from The Institutional Review Boards of the Jordan University (IRB approval no., 7/114/2018). All study subjects were informed of procedures and data collection prior to the start of study and signed a written consent form. Recruitment of study subjects took place between December 2018 and December 2019 at the diabetes and endocrinology clinics at King Abdullah University Hospital (KAUH), a tertiary hospital affiliated with Jordan University of Science and Technology in the northern part of Jordan. All research procedures were made to the Principle of Good Clinical Practice and the Declaration of Helsinki.

### Study population

A total of 400 diabetic patients, comprising 200 males and 200 females, were initially invited to participate in this study. Eligibility criteria required participants to be over 18 years of age with a prior diagnosis of T2DM. Exclusion criteria included individuals with type 1 diabetes mellitus (T1DM), pregnant women, and those with malignancies, Cushing's syndrome, thyroid dysfunction, patients who received blood transfusion in the last 4 months, and patients with hemoglobinopathies. Additionally, patients undergoing insulin therapy were excluded due to its potential impact on HOMA-IR calculations.

Out of the 400 invited, 240 participants consented and were pre-selected for the study. Cases were defined as individuals with uncontrolled T2DM (HbA1c >7%, n = 120), who were matched 1:1 with controls (HbA1c ≤7%, n = 120). Matching was based on age (±5 years), sex, body mass index (BMI) (±2 kg/m$^2$), and treatment modality (metformin monotherapy). Those patients had a confirmed diagnosis with T2DM by specialist endocrinologists according to the American Diabetes Association guidelines and had regular follow up at KAUH's outpatient endocrinology clinic.

The sample size for this study was calculated based on the estimated prevalence of T2DM in the adult population of Jordan [1,2]. With an adult population size of approximately 6 million and a T2DM prevalence of 15%, the sample size was determined using the standard formula for calculating sample size for population proportions:

$$n = Z^2 \times p \times (1-p)/E^2$$

where Z is the Z-score for a 95% confidence interval (1.96), p is the expected prevalence (0.15), and E is the margin of error set at 5% (0.05). Using this formula, the required sample size was calculated to be 196 subjects. However, the study included a total of 240 patients to ensure sufficient power and account for potential dropouts or missing data, maintaining a 95% confidence level and a 5% margin of error.

### Anthropometric measurements

Patients who met the eligibility criteria and agreed to participate in the study were interviewed by the attending physician during their visit to the clinic, and relevant information were obtained into a structured data collection sheet. Medical history, family history, height (cm), and weight (kg) of the patients were recorded during their visit. BMI was calculated based on the above measurements using the following equation: BMI = weight (kg)/height$^2$ (m$^2$). The age of the patients, their complete blood count (CBC), and their latest kidney function test results were recovered from the patients' electronic medical records.

## Blood sampling and handling

A certified phlebotomist withdrew two blood samples (5 ml each) by venipuncture from each participant after a 12-hour fast. One blood sample was collected into an ethylene-diaminete-tra-acetic acid (EDTA) tube (AFCO, Jordan) and then kept at 4˚C to be used for HbA1c measurement. The second blood sample was withdrawn into a plain tube containing a gel clot activator (AFCO, Jordan) and used to obtain serum following centrifugation for 5 minutes at 4000 × g. Serum samples were then aliquoted and frozen in -80˚C and subsequently used for the biochemical measurements of fasting blood glucose (FBS), total cholesterol, triglyceride, and insulin levels.

## HbA1c measurement

Around 5 ml of the withdrawn blood was stored in EDTA tubes then used to measure HbA1c levels at the laboratories of KAUH, using an automated analyzer system (Roche Diagnostics, Mannheim, Germany). Based on the HbA1c levels, patients were classified into controlled T2DM if HbA1c levels were lower than or equal to 7%, and uncontrolled T2DM if HbA1c levels were higher than 7%. We recruited a total of 120 patients with controlled T2DM and another 120 patients with uncontrolled T2DM. The two groups were matched by age, gender, and BMI.

## Laboratory analysis

Serum insulin levels were quantified using the Quantikine solid-phase sandwich ELISA kit (R&D Systems, Inc., St. Louis, MO, USA; cat. no. DINS00), with a sensitivity of 2.15 pmol/L and an assay range of 15.6–500 pmol/L. In brief, 100 μl of serum samples, diluted to fit the assay's detection range, were assayed in duplicate. A 100 μl standard solution was added to the wells of a 96-well plate pre-coated with a monoclonal antibody. After the required incubation, the plate was washed, and an enzyme-labeled antibody from the kit was applied, followed by substrate addition. The reaction was halted using a stop solution after color development. Optical density was measured at 450 nm using the ELx800 Microplate Reader (BioTek Instruments, Winooski, VT, USA). Additionally, glucose, total cholesterol, and triglyceride levels were analyzed at the KAUH laboratories using the cobas® modular analyzer system (Roche Diagnostics, Mannheim, Germany). Glucose levels were measured by the glucose oxidase method, while total cholesterol and triglycerides were quantified using enzymatic colorimetric assays. Insulin resistance (HOMA-IR) was calculated as follows: fasting serum insulin (μU/L) x fasting serum glucose (nmol/L) / 22.5 [26].

## Definitions of anemia and CKD

Anemia is defined according to the World Health Organization (WHO) standards as hemoglobin (Hb) levels <12.0 g/dL in non-pregnant women and <13.0 g/dL in men [27]. Kidney function was assessed using glomerular filtration rate (GFR) as estimated by the four-variable Modification of Diet in Renal Dis-ease (MDRD) study equation as follows: eGFR = 186.3 × (serum creatinine)-1.154 × (age) -0.203 × 1.212 (if black) × 0.742 (if female) [28]. GFR was expressed in ml/min/1.73 m2, and patients were considered to have chronic kidney disease when the estimated GFR (eGFR) was less than 60 mL/min/1.73 m2 according to the Kidney Disease Outcomes and Quality Initiative (KDIGO) guidelines [29].

## Statistical analysis

Statistical analyses were performed using SPSS (Systat Soft-ware, San Jose, CA, USA). One-way analysis of variance (ANOVA) followed by multiple pairwise comparison using the

Holm-Sidak method to test for significant differences in age, sex, BMI, WC, and serum levels of total cholesterol, triglyceride, and glucose between controlled and uncontrolled T2DM subjects. Chi-square test was used to assess the association between categorical variables (Gender, DM status, anemia, and CKD). A binomial logistic regression was performed to ascertain the effects of age, gender, CKD and DM status and the likelihood that participants have anemia. Moreover, the binomial logistic regression was performed to ascertain the effects of age, gender, anemia, and DM status the likelihood that participants have CKD. Furthermore, the variables in the binomial logistic regression (age, anemia, CKD) were analyzed based on gender, to ascertain their effect on the glycemic status. A P-value of <0.05 was used as a cut-off for significance. The logistic regression analysis was performed to derive the odds ratios (OR) and confidence intervals (CI). All data are presented as mean values ± standard deviation.

## Results

### Subject characteristics

During the study period, a total of 240 eligible subjects were recruited into the study. Those included 57 males and 63 females with controlled T2DM, and 57 males and 63 females with uncontrolled T2DM, matched by age and gender and BMI. No significant differences in age or BMI were observed between groups (P>0.1). The range of age was 40–90 years with a median of 60 years (Table 1).

### Biochemical profile of study population

To evaluate the biochemical profile of study subjects, one-way ANOVA was performed to compare parameters between groups, and the data are presented as mean ± SD. The biochemical profile of the subjects showed that uncontrolled T2DM subjects had significantly higher levels of FBS, higher HbA1c percentage, higher serum insulin levels, higher HOMA-IR score and a tendency for higher serum triglycerides, with no difference in cholesterol concentration between study groups, (Table 1).

### CBC parameters and kidney function test in T2DM patients

To evaluate the CBC parameters between study groups, their electronic medical records were reviewed, and their latest CBC data and kidney function test results were retrieved. One-way ANO-VA showed a significant difference between study groups in hemoglobin (Hgb) levels, red blood cells (RBC) count, red cell distribution width (RDW), mean cell hemoglobin (MCH), hematocrit (HCT) and mean cell volume (MCV). However, the post-hoc multiple comparison test showed no significant differences within male or female groups. The one-way ANOVA showed a significant difference between study groups in serum sodium, creatinine, urea levels and eGFR. The post-hoc multiple comparison showed that female patients with uncontrolled T2DM had significantly lower Na+ levels com-pared to controlled female patients (Table 1).

### Prevalence of anemia among T2DM patients

A chi-square test for association was conducted between anemia and status of glycemic control. Eighty-one patients out of 239 included in the study were anemic, therefore the overall prevalence of anemia among study subjects was 33.9%. The prevalence of anemia among controlled T2DM patients was 27.5%, and 40.0% among uncontrolled T2DM patients (OR: 2.14; 95% CI: 1.236 to 3.714, P = 0.0062). Importantly, among females, the prevalence of anemia among controlled T2DM patients was 28.6%, and 48.4% among uncontrolled T2DM patients

**Table 1. Baseline variables of the study subjects.**

| | Male | | Female | | |
|---|---|---|---|---|---|
| | Controlled T2DM (n = 57) | Uncontrolled T2DM (n = 57) | Controlled T2DM (n = 63) | Uncontrolled T2DM (n = 63) | P-Value[1] |
| Age(mean±SD) | 61.947±10.945 | 59.07±10.166 | 59.365±10.044 | 61.828±10.745 | 0.274 |
| BMI (mean±SD) | 28.947±5.016 | 29.358±4.51 | 31.188±5.694 | 30.912±5.391 | 0.161 |
| FBG (mg/dL) | 142.53±39.004 | 205.452±76.179* | 136.357±37.119 | 207.301±63.816* | **<0.001** |
| Insulin (pmol/L) | 30.808±25.14 | 41.076±39.151 | 25.707±16 | 41.408±41.307* | **0.015** |
| HBA1c (%) | 6.284±0.495 | 9.005±1.292* | 6.241±0.416 | 8.814±1.54* | **<0.001** |
| Cholesterol (mg/dL) | 199.361±53.713 | 204.376±59.391 | 215.938±59.549 | 216.492±50.945 | 0.249 |
| Triglycerides (mg/dL) | 160.201±102.069 | 192.59±145.108 | 136.957±75.664 | 174.591±126.312 | 0.058 |
| HOMA-IR | 11.281±10.829 | 20.785±21.718* | 8.83±6.123 | 21.286±19.82* | **<0.001** |
| Hgb (g/dl) | 13.958±2.306 | 13.58±2.023 | 12.471±1.508 | 12.103±1.566 | **<0.001** |
| RBC count (10^6*mm3) | 4.767±0.781 | 4.789±0.65 | 4.592±0.456 | 4.469±0.614 | **0.017** |
| HCT (%) | 41.415±6.573 | 40.449±5.512 | 37.716±4.111 | 36.375±4.396 | **<0.001** |
| MCV (fl) | 85.929±8.935 | 84.727±7.149 | 82.634±7.79 | 81.667±7.157 | **0.013** |
| MCH (pg) | 29.356±2.749 | 28.457±3.101 | 27.76±3.729 | 27.163±3.062 | **0.002** |
| RDW (%) | 16.538±16.738 | 14.611±1.961 | 14.818±1.904 | 14.934±1.781 | 0.585 |
| Na+ (mEq/L) | 139.768±4.336 | 138.786±3.726 | 141.061±3.448 | 139.141±2.949* | **0.004** |
| K+ (mmol/L) | 4.704±0.461 | 4.653±0.572 | 4.592±0.332 | 4.786±0.532 | 0.148 |
| Urea (mmol/L) | 9.023±13.425 | 8.213±5.918 | 4.915±2.254 | 7.273±6.915 | **0.037** |
| Creatinine (µmol/L) | 119.43±127.167 | 119.828±68.911 | 70.758±25.195 | 86.417±63.571 | **<0.00** |
| eGFR | 82.362±31.525 | 71.675±28.127 | 87.002±25.941 | 81±31.61 | **0.043** |

[1] The p-values were calculated using the one-way ANOVA.

* indicates a significant difference where p-value is <0.05

** indicates a significant difference where p-value is <0.001. The data are presented as the mean ± standard deviation. Abbreviations: BMI, body mass index; WC, waist circumference; HbA1c, glycated hemoglobin; FBS, fasting blood glucose, HOMA-IR, HgB, heamoglobin, RBC, red blood cells, HCT, hematocrit, MCV, mean cell volume, MCH, mean cell hemoglobin, RDW, red cell distribution width, eGFR, estimated glomerular filtration rate.

**Table 2. Prevalence of anemia and CKD among study subjects.**

**A.** Prevalence of Anemia among study subjects.

| | Controlled T2DM | Uncontrolled T2DM | P-Value | OR | 95% CI |
|---|---|---|---|---|---|
| Male (%, n) | 26.3% (15) | 29.8% (17) | 0.6335 | 1.221 | 0.5377 to 2.771 |
| Female (%, n) | 28.6% (18) | 48.4% (31) | 0.0255* | 2.296 | 1.100 to 4.792 |
| Overall (%, n) | 27.5% (32) | 40.0% (49) | 0.0062* | 2.143 | 1.236 to 3.714 |

**B.** Prevalence of CKD among study subjects.

| | Controlled T2DM | Uncontrolled T2DM | P-Value | OR | 95% CI |
|---|---|---|---|---|---|
| Male (%, n) | 24.6% (14) | 35.1% (20) | 0.1530 | 1.660 | 0.7370 to 3.740 |
| Female (%, n) | 17.5% (11) | 25.0% (16) | 0.3865 | 1.576 | 0.6718 to 3.631 |
| Overall (%, n) | 20.8% (25) | 30.0% (36) | 0.0742 | 1.609 | 0.8937 to 2.898 |

Note: P-values were calculated using the Pearson's chi-squared test of association. CKD, chronic kidney disease, T2DM, type 2 diabetes mellitus; %, percentage, n, number, OR, odds ratio, CI, confidence intervals.

(OR: 2.296; 95% CI: 1.100 to 4.792, P = 0.0255). However, among males the prevalence of anemia among controlled T2DM patients was 26.3% and 29.8% among uncontrolled T2DM patients (OR: 1.221; 95% CI: 0.5377 to 2.771, P = 0.6335) (Table 2A).

## Prevalence of CKD among T2DM patients

A chi-square test for association was conducted between CKD and status of glycemic control. The overall prevalence of CKD among study subjects was 25.5%. The prevalence of CKD among controlled T2DM patients was 20.8%, and 30.0% among uncontrolled T2DM patients (OR: 1.61; 95% CI: 0.8937 to 2.898 P = 0.074). Among females, the prevalence of CKD among con-trolled T2DM patients was 17.5%, and 25.0% among uncontrolled T2DM patients (OR: 1.58; 95% CI: 0.6718 to 3.631, P = 0.39). On the other hand, among males, the prevalence of CKD among controlled T2DM patients was 24.6% and 35.1% among uncontrolled T2DM patients (OR: 1.66; 95% CI: 0.7370 to 3.740, P = 0.15) (Table 2B).

## Association between gender, CKD and DM status the likelihood that participants have anemia

A binomial logistic regression was performed to ascertain the effects of gender, CKD and glycemic status the likelihood that participants have anemia. In the multivariate model, age was positively associated with development of anemia among study subjects (OR: 1.06; 95% CI: 1.023 to 1.090, P = 0.001). Moreover, females had 2.145 higher odds of exhibiting anemia compared to male patients (OR: 2.145; 95% CI: 1.140 to 4.019, P = 0.018). Furthermore, presence of CKD increases the likelihood of getting anemia in diabetic patients (OR: 6.425; 95% CI: 2.521 to 9.930, P = 0.000) (Table 3A). In the univariate model, age, uncontrolled T2DM and CKD were associated with increased likelihood of development of anemia, P<0.05 (Table 3B).

## Association between gender, anemia, and DM status the likelihood that participants have CDK

A binomial logistic regression was performed to ascertain the effects of gender, anemia, and glycemic status the likelihood that participants have CKD. In the multivariate model, age was

**Table 3. Binomial logistic regression predicting likelihood of anemia based on age, gender, glycemic status and CKD.**

**A.** *(Multivariate analysis).*

|  | B | SE | Wald | *df* | *P* | OR | 95% CI for OR | |
|---|---|---|---|---|---|---|---|---|
|  |  |  |  |  |  |  | Lower | Upper |
| Age | 0.054 | 0.016 | 11.173 | 1 | 0.001 | 1.056 | 1.023 | 1.090 |
| Gender | 0.761 | 0.321 | 5.605 | 1 | 0.018 | 2.140 | 1.140 | 4.019 |
| Glycemic status | 0.516 | 0.310 | 2.761 | 1 | 0.097 | 1.675 | 0.911 | 3.079 |
| CKD | 1.610 | 0.350 | 21.192 | 1 | 0.000 | 5.003 | 2.521 | 9.930 |
| Constant | -5.149 | 1.066 | 23.338 | 1 | 0.000 | 0.006 |  |  |

**B.** *(Univariate analysis).*

|  | B | SE | Wald | *df* | *P* | OR | 95% CI for OR | |
|---|---|---|---|---|---|---|---|---|
|  |  |  |  |  |  |  | Lower | Upper |
| Age | 0.065 | 0.015 | 19.798 | 1 | 0.000 | 1.068 | 1.037 | 1.099 |
| Gender | 0.477 | 0.277 | 2.952 | 1 | 0.086 | 1.611 | 0.935 | 2.775 |
| Glycemic status | 0.552 | 0.277 | 3.980 | 1 | 0.046 | 1.737 | 1.010 | 2.989 |
| CKD | 1.748 | 0.320 | 29.863 | 1 | 0.000 | 5.740 | 3.067 | 10.743 |

*Note*: Gender is for females compared to males.

**Table 4. Binomial logistic regression predicting likelihood of CKD based on age, gender, DM status and anemia.**

**A.** *(Multivariate analysis).*

| | B | SE | Wald | *df* | P | OR | 95% CI for OR | |
|---|---|---|---|---|---|---|---|---|
| | | | | | | | Lower | Upper |
| Age | 0.045 | 0.017 | 7.183 | 1 | 0.007 | 1.046 | 1.012 | 1.081 |
| Gender | -0.755 | 0.342 | 4.880 | 1 | 0.027 | 0.470 | 0.240 | 0.918 |
| Glycemic status | 0.371 | 0.337 | 1.209 | 1 | 0.271 | 1.449 | 0.748 | 2.806 |
| Anemia | 1.584 | 0.352 | 20.272 | 1 | 0.000 | 4.875 | 2.446 | 9.716 |
| Constant | -4.357 | 1.078 | 16.347 | 1 | 0.000 | 0.013 | | |

**B.** *(Univariate analysis).*

| | B | SE | Wald | *df* | P | OR | 95% CI for OR | |
|---|---|---|---|---|---|---|---|---|
| | | | | | | | Lower | Upper |
| Age | 0.064 | 0.015 | 17.239 | 1 | 0.000 | 1.066 | 1.035 | 1.099 |
| Gender | -0.456 | 0.299 | 2.333 | 1 | 0.127 | 0.634 | 0.353 | 1.138 |
| Glycemic status | 0.477 | 0.301 | 2.520 | 1 | 0.112 | 1.611 | 0.894 | 2.904 |
| Anemia | 1.748 | 0.320 | 29.863 | 1 | 0.000 | 5.740 | 3.067 | 10.743 |

*Note*: Gender is for females compared to males.

positively associated with development of CKD among study subjects (OR: 1.046; 95% CI: 1.012 to 1.081, P = 0.007). On the other hand, female gender was negatively associated with development of CKD (OR: 0.47; 95% CI: 0.240 to 0.918, P = 0.027), and existence of anemia increased the likelihood of developing CKD (OR: 4.875; 95% CI: 2.446 to 9.716, P<0.001) (Table 4A). In the univariate model, age and anemia were positively associated with increased likelihood of development of CKD, P<0.001 (Table 4B).

## Female T2DM patients are more likely to have uncontrolled T2DM and anemia

To evaluate the association between age, anemia and CDK and likelihood that partici-pants have uncontrolled T2DM, a binary logistic regression was performed to analyze these variables based on gender. Among the 3 variables of interest that were included in the model, age was negatively associated with uncontrolled diabetes among men (OR: 0.954; 95% CI: 0.915 to 0.994, P = 0.025), while anemia was positively associated with uncontrolled diabetes among females. (OR: 2.359, 95% CI: 1.084 to 5.133, P = 0.3) (Table 5).

## Discussion

The occurrence of anemia and renal insufficiency are well known co-morbidities in patients with T2DM. This study explored the effect of poor glycemic control on the prevalence of ane-mia and CKD among T2DM patients in Jordan using an observational case-control, age-, gen-der-, and BMI-matched study design. Here we report that the overall prevalence of anemia among T2DM patients was 33.9%; where controlled T2DM patients had a prevalence of 27.5%, while poor glycemic control is associated with a significant increase in its prevalence to 40%. The prevalence of anemia in this study (33.9%) is greater than that reported prevalence of anemia in the general adult population in Jordan (4.9% in males and 19.3% in non-pregnant females) [30–32], adding to the body of evidence that diabetic patients are at greater risk of developing anemia than non-diabetic individuals. Consistent with previous research in differ-ent populations, the binary logistic regression showed that lack of glycemic control is a signifi-cant predictor of development of anemia in T2DM patients in the univariate model, however,

**Table 5. Factors influencing development of uncontrolled T2DM based on gender.**

| Variables in the Equation | | | | | | | | | | |
|---|---|---|---|---|---|---|---|---|---|---|
| Gender | | | B | SE | Wald | *df* | *P* | OR | 95% CI for OR | |
| | | | | | | | | | Lower | Upper |
| Male | Step 1[a] | Age | -0.047 | 0.021 | 5.016 | 1 | 0.025 | 0.954 | 0.915 | 0.994 |
| | | CKD | 0.799 | 0.481 | 2.757 | 1 | 0.097 | 2.224 | 0.866 | 5.712 |
| | | Anemia | 0.552 | 0.538 | 1.050 | 1 | 0.305 | 1.736 | 0.605 | 4.984 |
| | | Constant | 2.468 | 1.229 | 4.034 | 1 | 0.045 | 11.796 | | |
| Female | Step 1[a] | Age | 0.019 | 0.019 | 0.978 | 1 | 0.323 | 1.019 | 0.982 | 1.057 |
| | | CKD | -0.161 | 0.476 | 0.115 | 1 | 0.734 | 0.851 | 0.335 | 2.162 |
| | | Anemia | 0.858 | 0.397 | 4.683 | 1 | 0.030 | 2.359 | 1.084 | 5.133 |
| | | Constant | -1.424 | 1.129 | 1.592 | 1 | 0.207 | 0.241 | | |

a. Variable(s) entered on step 1: Age, CKD (Chronic Kidney Disease), Anemia.

after adjusting for age, gender and CKD, lack of glycemic control is not significantly associated with development of anemia.

In spite of the reported a strong correlation between anemia and diabetes in epidemiological studies in various countries, the prevalence of anemia among diabetics shows wide variation in different populations and ethnicities, and often remains un-detected and undertreated. For instance, its prevalence was 11% in Korea [33], 22.8% in China [34], 23% in Australia [10], 29.3% in Oman [35], 31.12% in Ethiopia [36], 29.7% in Kuwait [37], 35% in African countries [38], 41.4% in Cameroon [39], 69.2% in Nigeria [40], and 31.7% in Malaysia [41]. This variation is largely attributed to several factors such as the socioeconomical status, geographical location and altitude, glycemic control, prevalence of infectious diseases and nutritional deficiencies [38–40]. Notably, advanced age, concurrent underlying co-morbidities, the existence of DM vascular complications, including renal impairment, macroalbuminuria, inadequate glycemic management, and duration of DM > 5 years, were the major risk factors for anemia in DM [13,14].

The etiology of anemia among DM patients is complex and multifactorial. Reduced hemoglobin (Hb) level indicative of anemia in diabetic patients is correlated with increased risk of hospitalization, mortality, and rapid deterioration in the glomerular filtration rate (GFR) and renal function due to renal ischemia and loss of functional nephrons [13,42]. The renal microcirculation is prone to damage in diabetes in the presence of combination of anemia and CKD. Several factors have been implicated in the earlier onset of anemia in diabetic patients, such as glucotoxicity, systemic inflammation, inhibition of erythropoietin release, damage to the renal interstitium, sympathetic denervation of the kidneys, and altered iron metabolism [31]. Anemia re-mains unidentified and largely under-treated in about quarter of T2DM patients probably due to the fact that both anemia and T2DM share similar symptoms like pale skin, chest pain, numbness or coldness in the extremities, shortness of breath and headache [43]. The number of patients suffering from concomitant diabetes and renal insufficiency is expected to double over the next decade, causing a significant increase in mortality and morbidity of the disease and increased burden on the healthcare system [44].

Importantly, in the current study, lack of glycemic control in female patients was associated with a significant increase in the prevalence of anemia compared to male patients, and the adjusted binary logistic regression showed that females had 2.14-fold increase in the likelihood of developing anemia compared to male patients, suggesting that female gender is an

important risk factor for development of anemia in diabetic individuals in Jordan. However, previous research reported that the effect of gender on development of anemia among diabetic patients varies in different populations. Past studies from Oman [36], Kuwait [37] and Malaysia [41] reported that anemia develop more frequently in female T2DM patients. On the other hand, studies from Ethiopia [36], China [34], United Kingdom [45] and Korea [33] reported higher prevalence of anemia among male T2DM patients compared to diabetic women. However, this study found an association between anemia and T2DM that differs between males and females. This finding is consistent with previous studies that have explored the relationship between iron deficiency anemia and HbA$_{1c}$ levels in DM, as well as the connection between anemia and kidney function in diabetic patients [33]. The possible reason for higher prevalence of anemia in females might be due to hormonal differences between men and women, health disparities, gender disparities, poor nutrition, illiteracy, and lack of women empowerment. Therefore, further research is needed to investigate the etiology of anemia in Jordanian populations with diabetes.

The binary logistic regression results showed that increased age among male T2DM patients is associated with reduced likelihood of having uncontrolled diabetes. Interestingly, some studies reported interesting findings where older diabetic patients are more likely to have better glycemic control despite increased prevalence of complications compared to young diabetic patients [46,47]. This may be partially explained by increased adherence to blood glucose control medications at an older age, suggesting a need to improve glycemic control in younger patients. This research points out that, beside the biological differences between males and females, males are probably more educated about glycemic management and have better access to health care. Therefore, for female patients with T2DM, educational interventions such as implementation of health awareness programs in rural areas, provision of iron-rich foods, prescription of vitamins and iron supplements, and awareness of the complications of diabetes may be necessary.

Adequate glycemic control in diabetic patients decreases the incidence of diabetic complications and improves their quality of life [48]. Chronic hyperglycemia increases the expression of proinflammatory cytokines such as interleukin-6, which has antierythropoietic effects leading to development of anemia [49]. Moreover, the hyper-glycemic state and disordered endocrine milieu induces microvascular changes within the kidney leading to development of CKD [50]. This disease is characterized by hemodynamic alterations characterized by hyperfiltration at the early stage followed by a progressive decline of renal function, development of albuminuria, glomerular basement membrane thickening, mesangial matrix expansion, nodular glomerulosclerosis, and arteriolar hyalinosis [9]. Diabetic nephropathy is a major microvascular complication of DM and is a major cause of end-stage renal disease (ESRD) in western populations [50]. Moreover, CKD is an important risk factor for cardiovascular mortality in diabetic patients. According to epidemiological studies, sexual dimorphism exists in CKD incidence and outcomes [50]. In the current study, CKD was present in 20.8% of controlled, and 30% of uncontrolled T2DM patients. The binary logistic regression showed that CKD is a significant predictor of anemia in T2DM patients (OR: 5.0 CI: 2.52 to 9.93, p<0.001) in the adjusted model.

Although it has been reported that CKD affects more women than men, exacerbated progression of CKD and development of ESRD affects more men compared with women [1]. These differences are attributed primarily due to differences in sex hormones between men and women since CKD differences are less evident after the postmenopausal age [23]. Sex hormones have been implicated in modulating renal function during health and disease. Testosterone is known to mediate renal injury through induction of renal vasoconstriction,

inflammation, oxidative stress, and apoptosis, while estrogens is known to exert reno-protective effects by inducing renal vasodilation, anti-inflammatory and anti-fibrotic effects [23,50].

One of the strengths of this study is that it explores the relationship between chronic diseases like DM and CKD and the development of anemia in a sex-specific manner, which has not been investigated extensively in many developing countries and presents an important avenue for future research. Recent study reported that nearly 50% of T2DM Jordan have diabetic kidney disease [23,24] and to our knowledge, this research is the first to examine sex-based differences in factors that influence anemia and CKD among T2DM patients in Jordan. On the other hand, we acknowledge that there were some significant limitations in this study, including the absence of a control group of healthy nondiabetic individuals, the reliance on a single measurement of eGFR, and incomplete records of proteinuria for all patients. Additionally, since this study was conducted in a primary care setting, many patients with anemia were not further investigated, leaving their underlying conditions unknown. Therefore, further investigation is needed to determine possible causes of anemia including iron or vitamin B12 deficiencies in these patients.

## Conclusions

This study revealed a significant prevalence of anemia among Jordanian patients with T2DM, with women being more affected than men, and the condition was found to be associated with poor glycemic control. This emphasizes the need for healthcare providers to integrate anemia screening into regular diabetes management protocol to detect and manage anemia promptly and improve the overall quality of care and well-being of diabetic individuals, since early identification and treatment of anemia can potentially reduce the risk of complications and improve the health outcomes of patients with T2DM.

## Author Contributions

**Conceptualization:** Ahmed Al-Dwairi, Othman Al-Shboul, Doa'a G. F. Al-U'datt.

**Data curation:** Ahmed Al-Dwairi, Othman Al-Shboul, Alhakam Albtoush, Aysam Hweidi, Abdulaziz Alnemer.

**Formal analysis:** Ahmed Al-Dwairi, Rami Saadeh, Adi Khassawneh.

**Funding acquisition:** Ahmed Al-Dwairi.

**Investigation:** Ahmed Al-Dwairi, Doa'a G. F. Al-U'datt, Mohammad AlQudah, Adi Khassawneh, Mahmoud Alfaqih.

**Methodology:** Ahmed Al-Dwairi, Othman Al-Shboul, Rami Saadeh, Mohammad AlQudah, Alhakam Albtoush, Aysam Hweidi, Abdulaziz Alnemer.

**Project administration:** Ahmed Al-Dwairi.

**Resources:** Ahmed Al-Dwairi, Mahmoud Alfaqih.

**Software:** Mohammad AlQudah.

**Supervision:** Ahmed Al-Dwairi.

**Validation:** Doa'a G. F. Al-U'datt, Mahmoud Alfaqih.

**Writing – original draft:** Ahmed Al-Dwairi.

**Writing – review & editing:** Ahmed Al-Dwairi.

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
