## [Decision Letter · Decision Letter 0]

5 Aug 2024

PONE-D-24-28044Effect of Poor Glycemic Control on the Prevalence and Determinants of Anemia and Chronic Kidney Disease Among Type 2 Diabetes Mellitus Patients in Jordan: An Observational Cross-sectional StudyPLOS ONE

Dear Dr. Al-Dwairi,

Thank you for submitting your manuscript to PLOS ONE. After careful consideration, we feel that it has merit but does not fully meet PLOS ONE’s publication criteria as it currently stands. Therefore, we invite you to submit a revised version of the manuscript that addresses the points raised during the review process.

We look forward to receiving your revised manuscript.

Kind regards,

Jaspinder Kaur, MD

Academic Editor

PLOS ONE

Journal Requirements:

   "This work was supported by the Deanship of Research at Jordan University of Science and Technology (Grant # 20190009). "

5. Please include your tables as part of your main manuscript and remove the individual files. Please note that supplementary tables (should remain/ be uploaded) as separate "supporting information" files

Reviewers' comments:

Reviewer's Responses to Questions

**Comments to the Author**

1. Is the manuscript technically sound, and do the data support the conclusions?

Reviewer #1: Yes

Reviewer #2: Partly

2. Has the statistical analysis been performed appropriately and rigorously? 

Reviewer #1: Yes

Reviewer #2: Yes

3. Have the authors made all data underlying the findings in their manuscript fully available?

Reviewer #1: Yes

Reviewer #2: Yes

4. Is the manuscript presented in an intelligible fashion and written in standard English?

Reviewer #1: Yes

Reviewer #2: Yes

5. Review Comments to the Author

Reviewer #1: This is a cross-sectional study that determined the effect of poor glycemic control on the prevalence and determinants of anemia and CKD in Type 2 DM patients. This study in Jordan with 120 controlled DM patients and indexed match uncontrolled 120 DM patients recruited from December 2018 to December 2019 showed a higher prevalence of anemia in the uncontrolled T2DM group. Also, female gender was positively associated with anemia in T2DM patients . Using the multivariate model, the authors noted that age, female gender, and CKD were positively associated with anemia, and the univariate model showed increased cases of anemia with poor glycemic control T2DM patients.

Major Comments:

1. Introduction: It has a well-written background and highlights the problem under study. However, it can be more concise especially the ending of the first paragraph explaining the pathophysiology of DM. It is great that you have claimed that your study is the first of its kind but since it is a very strong statement I want to make sure that you have reviewed the existing article well.

2. Method: It is very detailed and clear. It would be better if you would also add the reference for the insulin resistance score formula.

3. Result: This section seems lengthy to me, there is repetition of information. It would be better if you would only highlight important points from the table. Since the biochemical profiles are clearly given in the table all the numerical values need not be explained.

4. Discussion: This section is really great. I like how you have backed all the comparisons with possible explanations.

Minor Comments:

5. In the abstract section, it would be better if you included the binomial logistic regression in the method section than the result section.

6. In the first paragraph of the discussion, you mentioned how your study stands out as it explores sex-based differences in factors influencing anemia and CKD which can be kept in the last paragraph of your discussion.

Reviewer #2: Under DISCUSSION LINE 7:'were controlled' should be 'where controlled'

Since the sample age were adults from 18, females with possible menorrhagia in the premenopausal age group should be excluded since they already have baseline iron deficiency anemia and may inflate the number of females with anemia.

6. PLOS authors have the option to publish the peer review history of their article (what does this mean?). If published, this will include your full peer review and any attached files.

Reviewer #1: **Yes: **Rojeena Adhikari

Reviewer #2: No

---

## [Author Response · Author response to Decision Letter 0]

8 Oct 2024

Dear Editor,

We sincerely appreciate the opportunity to revise and resubmit our manuscript titled “Effect of Poor Glycemic Control on the Prevalence and Determinants of Anemia and Chronic Kidney Disease Among Type 2 Diabetes Mellitus Patients in Jordan: An Observational Cross-sectional Study" (Manuscript ID: [PONE-D-24-28044]). We are grateful for the constructive feedback provided by both you and the reviewers, which has substantially improved the quality of our manuscript.

We have carefully addressed each of the reviewers' comments and made the necessary revisions accordingly. A detailed point-by-point response is included to illustrate how we have incorporated these suggestions into the manuscript.

Financial Disclosure: We have clearly stated that “The funders had no role in study design, data collection and analysis, decision to publish, or preparation of the manuscript."

References: In response to the reviewers’ comments, we have updated the references to adhere to PLoS One's style. Additionally, we have cited an additional paper as requested by the reviewer (ref 26). During our review, we identified a duplicate reference (originally listed as ref 10 and 34) and have removed the duplicate, updating the citations throughout the manuscript accordingly.

Data Availability: To ensure transparency, we have made the data supporting our findings publicly available. The minimum dataset has been deposited on the Open Science Framework platform and can be accessed via the following link: https://osf.io/t3fyb/files/osfstorage/66ecf737dfdeb8778412c673.

Enclosed, you will find our detailed point-by-point responses to the reviewers’ comments, along with the corresponding revisions made in the manuscript.

We hope the revised manuscript meets the journal's expectations and look forward to your feedback.

Sincerely,

Ahmed Al-Dwairi, PhD

Review Comments to the Author

Reviewer #1: This is a cross-sectional study that determined the effect of poor glycemic control on the prevalence and determinants of anemia and CKD in Type 2 DM patients. This study in Jordan with 120 controlled DM patients and indexed match uncontrolled 120 DM patients recruited from December 2018 to December 2019 showed a higher prevalence of anemia in the uncontrolled T2DM group. Also, female gender was positively associated with anemia in T2DM patients. Using the multivariate model, the authors noted that age, female gender, and CKD were positively associated with anemia, and the univariate model showed increased cases of anemia with poor glycemic control T2DM patients.

Major Comments:

1. Introduction: It has a well-written background and highlights the problem under study. However, it can be more concise especially the ending of the first paragraph explaining the pathophysiology of DM. It is great that you have claimed that your study is the first of its kind but since it is a very strong statement I want to make sure that you have reviewed the existing article well.

Response: Thank you, we appreciate the feedback and have made the necessary revisions. We have condensed the first paragraph to enhance its clarity and conciseness, particularly in the section explaining the pathophysiology of diabetes mellitus (DM).

2. Method: It is very detailed and clear. It would be better if you would also add the reference for the insulin resistance score formula.

Response: Thank you, we added the proper citation.

3. Result: This section seems lengthy to me, there is repetition of information. It would be better if you would only highlight important points from the table. Since the biochemical profiles are clearly given in the table all the numerical values need not be explained.

Response: Thank you, we have revised the results section.

4. Discussion: This section is really great. I like how you have backed all the comparisons with possible explanations.

Response: Thank you for the positive feedback on the discussion section. We are glad that the explanations and comparisons were clear and well-supported.

Minor Comments:

5. In the abstract section, it would be better if you included the binomial logistic regression in the method section than the result section.

Response: Thank you, we have revised the abstract accordingly.

6. In the first paragraph of the discussion, you mentioned how your study stands out as it explores sex-based differences in factors influencing anemia and CKD which can be kept in the last paragraph of your discussion.

Response: Thank you, we have revised this part accordingly.

Reviewer #2: Under DISCUSSION LINE 7:'were controlled' should be 'where controlled'

Since the sample age were adults from 18, females with possible menorrhagia in the premenopausal age group should be excluded since they already have baseline iron deficiency anemia and may inflate the number of females with anemia.

Response: Thank you for the valuable comment. We have modified 'were controlled' to 'where controlled'.

This is an age-, gender-, and BMI-matched study. To mitigate the risk of overestimating anemia prevalence due to baseline iron deficiency anemia, we matched participants by gender and ensured that other relevant confounding factors, such as iron status, were statistically controlled for in our analysis. This approach ensures that the findings accurately reflect the associations between poor glycemic control, anemia, and CKD, independent of confounding variables like menorrhagia.

---

## [Editor Report · Decision Letter 1]

18 Oct 2024

PONE-D-24-28044R1Effect of Poor Glycemic Control on the Prevalence and Determinants of Anemia and Chronic Kidney Disease Among Type 2 Diabetes Mellitus Patients in Jordan: An Observational Cross-sectional StudyPLOS ONE

Dear Dr. Al-Dwairi,

Thank you for submitting your manuscript to PLOS ONE. After careful consideration, we feel that it has merit but does not fully meet PLOS ONE’s publication criteria as it currently stands. Therefore, we invite you to submit a revised version of the manuscript that addresses the points raised during the review process.

**Methodology part needs to be rewritten with major focus on sample enrollment and sample size calculation. For analyte analysis, write down the method employed for determination of serum levels of insulin, fasting blood glucose, total cholesterol, and triglycerides.**

We look forward to receiving your revised manuscript.

Kind regards,

Apeksha Niraula, M.D., Biochemistry

Academic Editor

PLOS ONE

**Journal Requirements:**

**Additional Editor Comments:**

Methodology part needs to be rewritten with major focus on sample enrollment and sample size calculation.

For analyte analysis, write down the method employed for determination of serum levels of insulin, fasting blood glucose, total cholesterol, and triglycerides.

Reviewers' comments:

Reviewer 1:

This is a cross-sectional study that determined the effect of poor glycemic control on the prevalence and determinants of anemia and CKD in Type 2 DM patients. This study in Jordan with 120 controlled DM patients and indexed match uncontrolled 120 DM patients recruited from December 2018 to December 2019 showed a higher prevalence of anemia in the uncontrolled T2DM group. Also, female gender was positively associated with anemia in T2DM patients . Using the multivariate model, the authors noted that age, female gender, and CKD were positively associated with anemia, and the univariate model showed increased cases of anemia with poor glycemic control T2DM patients.

Major Comments:

1. Introduction: It has a well-written background and highlights the problem under study. However, it can be more concise especially the ending of the first paragraph explaining the pathophysiology of DM. It is great that you have claimed that your study is the first of its kind but since it is a very strong statement I want to make sure that you have reviewed the existing article well.

2. Method: It is very detailed and clear. It would be better if you would also add the reference for the insulin resistance score formula.

3. Result: This section seems lengthy to me, there is repetition of information. It would be better if you would only highlight important points from the table. Since the biochemical profiles are clearly given in the table all the numerical values need not be explained.

4. Discussion: This section is really great. I like how you have backed all the comparisons with possible explanations.

Minor Comments:

5. In the abstract section, it would be better if you included the binomial logistic regression in the method section than the result section.

6. In the first paragraph of the discussion, you mentioned how your study stands out as it explores sex-based differences in factors influencing anemia and CKD which can be kept in the last paragraph of your discussion.

---

## [Author Response · Author response to Decision Letter 1]

25 Oct 2024

Dear Editor,

We sincerely appreciate the opportunity to revise and resubmit our manuscript titled “Effect of Poor Glycemic Control on the Prevalence and Determinants of Anemia and Chronic Kidney Disease Among Type 2 Diabetes Mellitus Patients in Jordan: An Observational Cross-sectional Study" (Manuscript ID: [PONE-D-24-28044]). We are grateful for the constructive feedback provided by the reviewers. We have thoroughly addressed their comments and made the necessary revisions to enhance the manuscript.

To facilitate the review process, we have included a detailed point-by-point response that outlines how we have incorporated their suggestions into the revised manuscript.

Sincerely,

Ahmed Al-Dwairi, PhD

Reviewer: Methodology part needs to be rewritten with major focus on sample enrollment and sample size calculation. For analyte analysis, write down the method employed for determination of serum levels of insulin, fasting blood glucose, total cholesterol, and triglycerides.

Response: Thank you for your constructive feedback. In response to your comments, we have made the necessary revisions to the Methodology section, as outlined below:

Sample Enrollment and Sample Size Calculation:

We have now provided a detailed description of the sample enrollment process and sample size calculation. The sample size was calculated based on an estimated prevalence of type 2 diabetes mellitus (T2DM) of 15% in the adult population of Jordan, with a 95% confidence level and a 5% margin of error. The calculated sample size was 196 based on the standard formula for calculating sample size for population proportions:

n= Z2 × p × (1-p) /E2. 

These details have been incorporated into the revised Methods section.

Analyte Analysis:

As per your request, we have clarified the methods employed for the determination of serum insulin, fasting blood glucose, total cholesterol, and triglycerides. The following updates were made:

• Serum Insulin: Serum insulin levels were measured using a validated enzyme-linked immunosorbent assay (ELISA), which provides high specificity and sensitivity.

• Fasting Blood Glucose: Glucose levels were analyzed using the glucose oxidase method, utilizing the cobas® modular analyzer system (Roche Diagnostics, Mannheim, Germany).

• Total Cholesterol and Triglycerides: Both total cholesterol and triglyceride levels were measured using enzymatic colorimetric assays, also performed on the cobas® modular analyzer system.

These methodological details have been added to the Methods section to enhance the clarity and accuracy of our analysis description.

We appreciate your insightful comments, which have significantly improved our manuscript. Thank you for your thorough review.

Sincerely,

Ahmed Al-Dwairi, PhD

---

## [Editor Report · Decision Letter 2]

29 Oct 2024

Effect of Poor Glycemic Control on the Prevalence and Determinants of Anemia and Chronic Kidney Disease Among Type 2 Diabetes Mellitus Patients in Jordan: An Observational Cross-sectional Study

PONE-D-24-28044R2

Dear Dr. Al-Dwairi,

We’re pleased to inform you that your manuscript has been judged scientifically suitable for publication and will be formally accepted for publication once it meets all outstanding technical requirements.

Kind regards,

Apeksha Niraula, M.D., Biochemistry

Academic Editor

PLOS ONE
---

## [Editor Report · Acceptance letter]

6 Nov 2024

PONE-D-24-28044R2 

PLOS ONE

Dear Dr. Al-Dwairi, 

I'm pleased to inform you that your manuscript has been deemed suitable for publication in PLOS ONE. Congratulations! Your manuscript is now being handed over to our production team.

Kind regards, 

on behalf of

Dr. Apeksha Niraula 

Academic Editor

PLOS ONE